# Age-Dependent and Aβ-Induced Dynamic Changes in the Subcellular Localization of HMGB1 in Neurons and Microglia in the Brains of an Animal Model of Alzheimer’s Disease

**DOI:** 10.3390/cells13020189

**Published:** 2024-01-18

**Authors:** Song-I Seol, Dashdulam Davaanyam, Sang-A Oh, Eun-Hwa Lee, Pyung-Lim Han, Seung-Woo Kim, Ja-Kyeong Lee

**Affiliations:** 1Department of Anatomy, Inha University School of Medicine, Incheon 22212, Republic of Korea; ssie8878@naver.com (S.-I.S.); dashka0109255@gmail.com (D.D.); bbobbibbo2@naver.com (S.-A.O.); 2Department of Brain and Cognitive Sciences, Scranton College, Ewha Womans University, Seoul 03760, Republic of Korea; gonobplaypan@nate.com (E.-H.L.); plhan@ewha.ac.kr (P.-L.H.); 3Department of Chemistry and Nano Science, College of Natural Science, Ewha Womans University, Seoul 03760, Republic of Korea; 4Department of Biomedical Sciences, Inha University School of Medicine, Inchon 22212, Republic of Korea

**Keywords:** HMGB1, AD, Aβ, frontal cortex, aging

## Abstract

HMGB1 is a prototypical danger-associated molecular pattern (DAMP) molecule that co-localizes with amyloid beta (Aβ) in the brains of patients with Alzheimer’s disease. HMGB1 levels are significantly higher in the cerebrospinal fluid of patients. However, the cellular and subcellular distribution of HMGB1 in relation to the pathology of Alzheimer’s disease has not yet been studied in detail. Here, we investigated whether HMGB1 protein levels in brain tissue homogenates (frontal cortex and striatum) and sera from Tg-APP/PS1 mice, along with its cellular and subcellular localization in those regions, differed. Total HMGB1 levels were increased in the frontal cortices of aged wildtype (7.5 M) mice compared to young (3.5 M) mice, whereas total HMGB1 levels in the frontal cortices of Tg-APP/PS1 mice (7.5 M) were significantly lower than those in age-matched wildtype mice. In contrast, total serum HMGB1 levels were enhanced in aged wildtype (7.5 M) mice and Tg-APP/PS1 mice (7.5 M). Further analysis indicated that nuclear HMGB1 levels in the frontal cortices of Tg-APP/PS1 mice were significantly reduced compared to those in age-matched wildtype controls, and cytosolic HMGB1 levels were also significantly decreased. Triple-fluorescence immunohistochemical analysis indicated that HMGB1 appeared as a ring shape in the cytoplasm of most neurons and microglia in the frontal cortices of 9.5 M Tg-APP/PS1 mice, indicating that nuclear HMGB1 is reduced by aging and in Tg-APP/PS1 mice. Consistent with these observations, Aβ treatment of both primary cortical neuron and primary microglial cultures increased HMGB1 secretion in the media, in an Aβ-dose-dependent manner. Our results indicate that nuclear HMGB1 might be translocated from the nucleus to the cytoplasm in both neurons and microglia in the brains of Tg-APP/PS1 mice, and that it may subsequently be secreted extracellularly.

## 1. Introduction

Alzheimer’s disease (AD) is one the most common neurodegenerative diseases worldwide. The pathological features of AD include the abnormal deposition of extracellular amyloid β (Aβ) protein in senile plaques, along with progressive loss of memory and cognitive function [1]. Accumulating evidence suggests that the deposition of Aβ is a critical initial step in the pathogenesis of AD, which can induce pathological forms of Aβ fibrils that trigger a series of detrimental events and spread the disease [2]. In addition, reductions in Aβ clearance by microglia [3] and increases in oxidative stress [4] and neuroinflammatory processes [5] may promote the accumulation of toxic Aβ species. Moreover, many factors can increase the inflammatory components, including numerous danger-associated molecular patterns (DAMPs) that promote and accelerate AD’s pathogenesis [5,6].

HMGB1 is a highly conserved and ubiquitously expressed prototypical DAMP. It was initially reported as a nuclear protein that acts as a DNA chaperone and regulates DNA transcription, replication, and repair [7]. Subsequently, HMGB1 was found to be passively released from necrotic or injured cells and actively secreted by immune cells, such as monocytes and macrophages [8,9]. Extracellular HMGB1 regulates inflammatory and immune responses through cooperation with other factors, such as chemokines, growth factors, and cytokines [10,11]. There are two isoforms of HMGB1, depending on the redox state: disulfide HMGB1 and reduced HMGB1, which interact with TLR4 and CXCR4, respectively, and exert pro-inflammatory and chemotactic effects [12,13]. 

A critical role of HMGB1 in AD has been suggested in recent years. HMGB1 levels are significantly higher in the brains of AD patients compared to non-AD patients, and HMGB1 co-localizes with and stabilizes Aβ [14,15,16]. Furthermore, HMGB1 reduces Aβ phagocytosis by rat microglia [14], and the co-injection of Aβ42 and HMGB1 into the brains of rats reduces Aβ42 phagocytosis in the hippocampus [17]. In addition, intracerebroventricular injection of HMGB1 exacerbates Aβ42-induced long-term memory loss, and this was blocked by a TLR4 antagonist or in RAGE-deficient mice, suggesting the importance of both TLR4 and RAGE receptors in HMGB1- and Aβ42-induced pathogenesis [18]. In the brains of Tg2576 mice, Aβ induces HMGB1 secretion, and secreted HMGB1 promotes neuronal cell death [19]. Fujita et al. [16] demonstrated that subcutaneous injection of an anti-HMGB1 antibody potently inhibited neurite degeneration, even in the presence of Aβ plaques, and completely restored cognitive impairment in a Tg6799 mouse model of AD. The release of HMGB1 is increased in the brains and cerebrospinal fluid of patients with AD, as well as in elderly rodents [6,16,20]. 

In this study, we investigated the protein levels of HMGB1 and its cellular and subcellular localization in the frontal cerebral cortices and striata of Tg-APP/PS1 mice. We also investigated Aβ-mediated HMGB1 secretion by neurons and microglia. Tg-APP/PS1 mice express the human Swedish amyloid precursor protein (APPswe) and the ΔE9 presenilin 1 mutation (PSEN1dE9) under the control of the PDGF promoter [21,22]. We chose the Tg-APP/PS1 mouse line for the present study because it presents nearly all aspects of Aβ42-related pathologies known in animal models. 

## 2. Materials and Methods

### 2.1. Animals and Sample Preparation

The generation of Tg-APPswe/PS1dE9 mice (Tg-APP/PS1 mice), which carry the *human* Swedish *amyloid precursor protein* (APPswe) and the ΔE9 *presenilin 1 mutation* (PSEN1dE9), has been described previously [21,22]. The Tg-APPswe/PS1dE9 mice were crossed with C57BL6 × C3H hybrid mice, and the genetic hybrid was maintained. After weaning, male and female mice were separately housed (2–3 animals per cage) under a 12 h light/dark cycle (lights on at 7 a.m. and off at 7 p.m.) in a humidity (50~60%)- and temperature (22~24 °C)-controlled and specific pathogen-free (SPF) conditioned animal room. Genotyping of the Tg-APP/PS1 mice was determined by genomic PCR of tail biopsy samples using the following primers: 5′-CTAGGCCACAGAATTGAAAGATCT-3′ and 5′-GTAGGTGGAAATTCTAGCATCATCC-3′ for WT, 5′-AATAGAGAACGGCAGGAGCA-3′ and 5′-GCCATGAGGGCACTAATCAT-3′ for the PS1 gene, and 5′-AGGACTGACCACTCGACCAG-3′ and 5′-CGGGGGTCTAGTTCTGCAT-3′ for the APP gene. The Tg-APP/PS1 mice were maintained and handled in accordance with the Animal Care Guidelines of Ewha Womans University (IACUC 19-016).

### 2.2. Serum and Brain Tissue Sample Preparation

The mice were anesthetized with an intramuscular (i.m.) injection of a mixture of ketamine (100 mg/kg body weight) and xylazine hydrochloride (23.32 mg/kg body weight), and the frontal cortices and striata of the mice’s brains were isolated and frozen in liquid nitrogen for subsequent experiments. For serum samples, blood samples were collected through cardiac puncture using a 23 G syringe and left at room temperature for 30 min. The blood samples were centrifuged for 15 min at 2000× *g* and 4 °C. The supernatant was aliquoted and stored at −80 °C.

### 2.3. Immunoblotting

The mice were anesthetized with an intramuscular (i.m.) injection of a mixture of ketamine (100 mg/kg body weight) and xylazine hydrochloride (23.32 mg/kg body weight) and transcardially perfused with 0.9% saline. The brains were removed, and the frontal cortex and striatum were surgically obtained. Brain tissues were homogenized and lysed in RIPA buffer (0.25% sodium deoxycholate, 150 mM NaCl, 50 mM Tris-HCl (pH 7.4), 1% NP-40) containing a Complete Mini protease inhibitor cocktail tablet (Roche Diagnostics, Basel, Switzerland). The lysates were centrifuged at 12,000× *g* for 10 min at 4 °C, and the supernatants were loaded onto 12% SDS PAGE gels and separated at 100 V for 120 min. Separated proteins from the gels were transferred onto polyvinylidene fluoride membranes (PVDF; IPVH00010; Merck Millipore, Darmstadt, Germany) at 75 V for 150 min at 4 °C and blocked with 5% nonfat milk for 1 h. The primary antibodies used were as follows: anti-HMGB1 (ab18256; Abcam, Cambridge, UK), anti-α-tubulin (GTX112141; GeneTex, Irvine, CA, USA), and anti-laminB1 (ab133741; Abcam). The secondary antibodies (horseradish-peroxidase-conjugated secondary antibodies (AP132P; Merck Millipore, Darmstadt, Germany) were detected using a chemiluminescence kit (Merck Millipore). 

### 2.4. Enzyme-Linked Immunosorbent Assay (ELISA)

The levels of HMGB1 in the serum and cell lysates and culture media of primary cortical and primary microglial cultures were assayed using ELISA kits (Cusabio, Houston, TX, USA). For serum samples, whole blood was collected from Tg-APP/PS1 or age-matched Wt mice at 3.5 or 7.5 M of age, and it was allowed to clot by leaving it undisturbed for 20 min at room temperature. Clots were then removed by centrifuging the samples at 3000 rpm for 20 min in a cold microcentrifuge. The supernatants were immediately transferred to clean polypropylene tubes, and the concentrations were determined using ELISA kits. Time-course protein samples were prepared from cell lysates and culture media, rinsed with PBS, homogenized in 1 mL of PBS, and centrifuged for 5 min at 5000 rpm and −4 °C. The homogenates were assayed using ELISA kits (Cusabio, Houston, TX, USA).

### 2.5. Nuclear and Cytoplasmic Extract Preparation 

Nuclear and cytoplasmic extract preparation was performed as previously described [23]. Brain tissue blocks from the frontal cortices and striata were obtained surgically. They were then homogenized and lysed with solution A (0.5% TritionX-100, 0.5% NP-40, 10 mM HEPES (pH 7.9), 10 mM KCl, 0.1 mM EDTA, and 1 mM DTT) containing a Complete Mini protease inhibitor cocktail tablet (Roche Diagnostics, Basel, Switzerland), with 10 pumping cycles, using a 31 G syringe to collect cytoplasmic proteins. The lysates were centrifuged at 17,500× *g* for 10 min at 4 °C. The supernatant containing the cytoplasmic protein was stored at −80 °C. Pellets containing the nuclear protein were lysed with solution B (10% glycerol, 20 mM HEPES (pH 7.9), 0.4 M NaCl, 1 mM EDTA, and 1 mM DTT) containing a Complete Mini protease inhibitor cocktail (Roche Diagnostics). The lysates were centrifuged at 17,500× *g* for 10 min at 4 °C, and the supernatant was stored at −80 °C.

### 2.6. Immunohistochemistry

Brain tissues were fixed in 4% PFA, embedded in paraffin, and sectioned into 5 µm slices using a microtome. The sections were incubated with rabbit anti-HMGB1 antibody (Novus Biologicals, Littleton, CO, USA) overnight, followed by incubation with a secondary HRP-labeled antibody for 1 h at room temperature. Subsequently, the sections were counterstained with hematoxylin and eosin (H&E). Images were captured using an Olympus IX83 microscope, Tokyo, Japan.

### 2.7. Immunofluorescence Staining

Immunohistochemical analyses were performed as previously described [22,24]. Briefly, the mice were anesthetized with an intramuscular (i.m.) injection of a mixture of ketamine (100 mg/kg body weight) and xylazine hydrochloride (23.32 mg/kg body weight), and then they were transcardially perfused with 0.9% saline, followed by 4% paraformaldehyde in 0.1 M phosphate buffer (PBS; 137 mM NaCl, 2.7 mM KCl, 10 mM Na_2_HPO_4_, and 1.8 mM KH_2_PO_4_, pH 7.4). Their brains were surgically removed and incubated in 4% paraformaldehyde buffer at 4 °C for an additional 48 h. Then, they were coronally cut to a thickness of 40 μm using a vibratome (Leica VT 1000S, Leica instruments, Mussloch, Germany). Brain tissue sections were washed with PBS containing 0.1% Triton X-100 and blocked with a blocking solution (5% FBS, 5% horse serum, 2% BSA, and 0.1% Triton X-100 in PBS) for 1 h at room temperature. Primary antibodies were diluted 1:200 for anti-Iba1 (orb18542; Biorbyt, Cambridge, UK), anti-NeuN (NBP3-05554; Novus Biologicals, St. Charles, MO, USA), anti-HMGB1 (NBP-25148; Novus Biologicals), and anti-beta amyloid (700254; Invitrogen, Waltham, MA, USA). Brain tissues were incubated in diluted primary antibody solutions overnight at 4 °C. The sections were washed with PBS and incubated with Alexa Fluor 405-conjugated anti-rabbit IgG (1:200; Abcam), FITC-conjugated anti-mouse IgG (1:300; Merck Millipore), or Alexa Fluor 594-conjugated anti-goat IgG (1:200; Abcam) for 1 h at room temperature. The sections were mounted on slides using VECTASHIELD Antifade Mounting Solution without DAPI (Vector Laboratories, Newark, CA, USA) and examined under a Zeiss LSM 510 META microscope (Carl Zeiss Meditec AG, Jena, Germany).

### 2.8. Primary Neuron Cultures

Experiments were conducted in accordance with the Guide for the Care and Use of Laboratory Animals published by the National Institutes of Health (NIH, USA, 2011). The animal protocol used in this study was reviewed and approved by the INHA-IACUC (approval number INHA 20201103-734). All efforts were made to reduce the number of animals used and to minimize animal suffering. Briefly, mixed cortical cells were prepared from embryonic day 15.5 (E15.5) mouse cortices and cultured, as described previously [25]. Dissociated cortical cells were plated at a density of five hemispheres per 24-well poly-D-lysine (100 μg/mL)- and laminin (100 μg/mL)-coated plate (approximately 4 × 10^5^ cells/well). Cultures were maintained in MEM containing 5% FBS, 5% horse serum, 2 mM glutamine, and 21 mM glucose, without antibiotics. When astrocytes had reached confluence under neurons at day 7 in vitro (DIV 7), cytosine arabinofuranoside (ara-C) was added to a final concentration of 10 μM, and the culture was maintained for two days to halt microglial growth. FBS and glutamine were not supplemented after DIV 7. The cultures were used for experiments on DIV 12–14.

### 2.9. Primary Microglial Cultures

Primary microglial cultures were prepared as described previously [26]. Briefly, cells that dissociated from the cerebral hemispheres of 1-day-old postnatal rat brains (Sprague-Dawley strain) were seeded into 75 cm^2^ flasks at a density of 1.2 × 10^6^ cells/mL in Dulbecco’s modified Eagle’s medium (DMEM; Hyclone, Logan, UT, USA) containing 10% FBS and 1% penicillin/streptomycin (Gibco BRL, Gaithersburg, MD, USA). After culturing for 2 weeks, microglia were detached from the flasks by mild shaking and filtered using a 40 μm cell strainer (BD Falcon, Bedford, MA, USA) to remove astrocytes. After centrifugation (1000× *g*) for 5 min, the cells were resuspended in fresh DMEM containing 10% FBS and 1% penicillin/streptomycin. The cells were plated at a final density of 4 × 10^5^ cells/well in a 6-well culture plate, and the medium was changed to DMEM containing 10% FBS and 100X B27 supplement (Gibco) after 2 h.

### 2.10. Preparation of Peptides and Treatment of Amyloid Beta Oligomers

Amyloid beta (1–42) (Aβ, GenScript, Piscataway, NJ, USA) and scrambled amyloid beta (1–42) (scAβ, rPeptide, Watkinsville, GA, USA) were dissolved in PBS (pH 7.4) to 200 μM and stored at −80 °C until use. Oligomerization of Aβ was performed as described previously [27,28]. Briefly, the peptides were sonicated for 1 min using a 50/60 Hz sonicator immediately before use to remove any pre-aggregates. The solution was diluted in fresh treated medium and incubated with rotation for 6 h at 4 °C to generate amyloid beta oligomer formation. The medium containing oligomers was diluted to concentrations of 0.5, 1, or 3 μM and treated for 12 or 24 h.

### 2.11. Cell Viability Assay

Cell viability assays in primary neurons or microglia were performed using Cell Counting Kit-8 (Dojindo Molecular Technologies, Rockville, MD, USA) according to the manufacturer’s recommendations.

### 2.12. Nitric Oxide Assay

Primary microglia (1.5 × 10^5^ cells/well) were seeded into 24-well plates and, after 1 day, they were treated with or without synthesized amyloid beta oligomer, scrambled amyloid beta, or LPS (100 ng/mL) for 24 h. Then, 100 μL of conditioned medium was mixed with 100 μL of Griess reagent (0.5% sulfanilamide, 0.05% N-naphthalene-diamine-H-chloride, and 2.5% H_3_PO_4_) and incubated for 10 min at room temperature. NaNO_2_ was used as a standard to measure the NO_2_^−^ concentrations. The absorbance of the mixtures was measured at 550 nm using a microplate reader to measure nitric oxide production.

### 2.13. Statistical Analysis

Statistical analysis was performed using analysis of variance (ANOVA) followed by the Newman–Keuls test. The analyses were performed using PRISM software 5.0 (Graph Pad Software), and all data are presented as the means ± SEMs; *p*-values < 0.05 were accepted as statistically significant.

## 3. Results

### 3.1. HMGB1 Protein Levels in the Frontal Cortices, Striata, and Sera of Tg-APP/PS1 and Age-Matched Control Mice

Tg-APP/PS1 mice showed plaque deposition starting from approximately 6.5 M of age, and at 7.5 M of age more Aβ plaques were detected in the frontal cortex than in the striatum [22,29]. To examine the levels of HMGB1 in the frontal cortices and striata of Tg-APP/PS1 mice, immunoblot analyses were conducted with protein samples obtained from 3.5 or 7.5 M Tg-APP/PS1 mice and age-matched control mice. HMGB1 levels in the frontal cortices of Tg-APP/PS1 mice at 3.5 M of age were slightly higher than those in the wildtype control mice at 3.5 M, but the difference was not statistically significant. The HMGB1 levels in the frontal cortices of Tg-APP/PS1 mice at 7.5 M were significantly lower than those in age-matched wildtype control mice (Figure 1A,B). However, in the striatum, the HMGB1 levels in Tg-APP/PS1 mice and age-matched wildtype control mice were comparable at both 3.5 and 7.5 M (Figure 1C,D). Interestingly, the HMGB1 levels of wildtype control mice were significantly higher in aged animals (7.5 M) compared to those in mice at 3.5 M of age in both the frontal cortices and the striata (Figure 1A–D). The ELISA results showed that the HMGB1 levels in the serum were significantly higher in aged wildtype mice than those in young wildtype mice, and the HMGB1 levels in Tg-APP/PS1 mice at 7.5 M were significantly higher than those in aged wildtype mice (Figure 1E). Similar results were obtained from the immunoblot analysis (Appendix A). Collectively, these results suggest that HMGB1 levels change dynamically in wildtype and Tg-APP/PS1 mice with age. HMGB1 levels increased in the frontal cortex and striatum with age, but HMGB1 levels in the frontal cortices of Tg-APP/PS1 mice (7.5 M) were lower than those of wildtype mice; however, the levels in the striatum were comparable to those of wildtype mice. 

### 3.2. Decreased HMGB1 Levels in the Nuclei in the Brains of Tg-APP/PS1 Mice

The decrease in HMGB1 levels in the frontal cortex and its accumulation in the sera of Tg-APP/PS1 mice at 7.5 M prompted us to investigate the possible redistribution of HMGB1 at the subcellular level. Immunoblot analysis indicated that the levels of nuclear HMGB1 were significantly lower in the frontal cortices of Tg-APP/PS1 mice at 7.5 M compared to those of age-matched wildtype mice, and the cytosolic HMGB1 levels in Tg-APP/PS1 mice were also lower than those in age-matched wildtype mice (Figure 2A,B). In contrast, both nuclear and cytosolic HMGB1 levels in the striata of Tg-APP/PS1 mice (7.5 M) were comparable to those of age-matched wildtype controls (Figure 2C,D). To further investigate the redistribution of HMGB1 at the subcellular level in the frontal cortex, we performed immunohistochemistry using an anti-HMGB1 antibody. HMGB1 immunoreactivity was significantly decreased in the frontal cortices of 7.5 M Tg mice, with a marked alteration in the subcellular localization of HMGB1, which was detected primarily in the cytoplasm (Figure 2E,F, double arrowheads). Collectively, these results indicate that HMGB1 levels were reduced in both the nuclei and cytoplasm in the frontal cortices, whereas they were not significantly different in the striata of Tg-APP/PS1 mice at 7.5 M. 

### 3.3. Distribution of HMGB1 in the Nuclei and Cytoplasm of Neurons and Microglia in the Brains of Tg-APP/PS1 Mice at 3.5 M

To examine the cellular and subcellular distribution of HMGB1 in Tg-APP/PS1 mice, immunofluorescence staining was performed using anti-Aβ, anti-HMGB1, and anti-Iba1 (a microglial marker) antibodies. No Aβ staining was detected in the frontal cortices of wildtype mice at 3.5 M of age (Figure 3A). Iba1-positive microglia appeared small and ramified, suggesting that they were in a resting state, and HMGB1 immunoreactivity was localized in the nucleus (arrows in Figure 3B,C). HMGB1 immunoreactivity was also detected in Iba1-negative cells, and the majority of the cells appeared to be neurons based on their large cell size, nucleus, and shape, wherein HMGB1 immunoreactivity was detected in both the nucleus and cytoplasm (arrowheads in Figure 3B,C). Immunofluorescence staining using anti-Aβ, anti-HMGB1, and anti-NeuN (a neuronal marker) antibodies further confirmed the localization of HMGB1 in both the nuclei and cytoplasm of neurons in the frontal cortices (arrowheads in Figure 3E,F). Notably, numerous NeuN-negative/HMGB1-positive cells were detected in wildtype mice at 3.5 M of age (arrows in Figure 3E,F). In the frontal cortices of Tg-APP/PS1 mice at 3.5 M of age, Iba1 staining showed that most microglia were in a resting state, and HMGB1 was detected mainly in the nuclei (arrows in Figure 3G–I), with cytosolic localization in some Iba1-positive cells (double arrow in Figure 3H). In Iba1-negative/HMGB1-positive cells, HMGB1 was detected in both the nucleus and the cytoplasm (arrowheads in Figure 3H,I) in some cells, whereas it was only detected in the cytoplasm in others (double arrowhead in Figure 3I). Triple-immunofluorescence staining with anti-Aβ, anti-HMGB1, and anti-NeuN antibodies further confirmed the localization of HMGB1 in the nuclei and cytoplasm of neurons (arrowhead in Figure 3H). These results indicate that HMGB1 was detected in both the nuclei and cytoplasm of most, but not all, microglia and neurons in the frontal cortices of Tg-APP/PS1 mice. 

### 3.4. Changes in the Distribution of HMGB1 in the Nuclei of Neurons and Microglia in the Frontal Cortices of Tg-APP/PS1 Mice at 9.5 M of Age

In the frontal cortices of wildtype mice at 9.5 M of age, Aβ staining was not detected (Figure 4A,C). Triple-immunofluorescence staining with anti-Aβ, anti-HMGB1, and anti-Iba1 antibodies showed that most Iba1-positive cells were in a resting state and exhibited an extensive ramified morphology (Figure 4B). Moreover, HMGB1 was detected in both the nuclei (arrows in Figure 4B) and the cytoplasm (double arrow in Figure 4B) of microglia. Among Iba1-negative/HMGB1-positive cells, the majority appeared to be neurons, and HMGB1 was detected in both the nuclei and the cytoplasm (arrowheads in Figure 4B). Triple-immunofluorescence staining with anti-Aβ, anti-HMGB1, and anti-NeuN antibodies further confirmed the nuclear and cytoplasmic localization of HMGB1 in the neurons (arrowheads in Figure 4D); however, cytoplasmic localization of HMGB1 was detected more often in NeuN-negative cells (double arrows in Figure 4D). In the frontal cortices of Tg-APP/PS1 mice at 9.5 M of age, Aβ accumulation was prominent and distributed throughout (Figure 4E–H). The numbers of microglia increased and they clustered around Aβ plaques (Figure 4E,F). Overall, HMGB1 immunoreactivity was enhanced in Iba1-positive cells and was mainly detected in the cytoplasm (double arrows in Figure 4(F1–F4)). In Iba1-negative cells (the majority of which appeared to be neurons), HMGB1 was detected mainly in the cytoplasm, in a ring shape (double arrowhead in Figure 4(F1–F3)), and only some showed nuclear localization (arrowheads in Figure 4(F1–F3)). Consistent with this observation, triple-immunofluorescence staining with anti-Aβ, anti-HMGB1, and anti-NeuN antibodies further showed the cytoplasmic localization of HMGB1 in neurons (double arrowhead in Figure 4(H1,H2)) and NeuN-negative cells (double arrows in Figure 4(H1,H2)). To quantify the nuclear and cytoplasmic localization of HMGB1 in neurons in the frontal cortices of 9.5 M Tg mice, we measured the fluorescence intensity of HMGB1 co-localized with DAPI (nuclear), or not (cytoplasmic), in neurons. This analysis revealed a significant decrease in nuclear HMGB1 (arrowheads), while cytoplasmic HMGB1 (double arrowheads) showed a slight but significant decrease (Figure 4I,J). Collectively, these results indicate that, in the frontal cortices of Tg-APP/PS1 mice at 9.5 M of age, HMGB1 was mainly located in the cytoplasm of both neurons and microglia.

### 3.5. Distribution of HMGB1 in the Nuclei and Cytoplasm of Neurons and Microglia in the Striata of Tg-APP/PS1 Mice at 9.5 M of Age 

Aβ staining was not detected in the striata of wildtype mice at 3.5 M of age (Figure 5A,B). Triple-immunofluorescence staining with anti-Aβ, anti-HMGB1, and anti-Iba1 antibodies showed that most Iba1-positive cells were in the resting state, wherein HMGB1 was detected in both the nucleus (arrows in Figure 5B) and the cytoplasm (double arrow in Figure 5B). In Iba1-negative/HMGB1-positive cells, the majority of which appeared to be neurons, HMGB1 was detected in both the nucleus and the cytoplasm (arrowheads in Figure 5B). A similar staining pattern was detected in the striata of Tg-APP/PS1 mice at 3.5 M of age (Figure 5C,D). At 9.5 M of age, Aβ accumulation was not detected in wildtype mice (Figure 5E,F); however, it was weakly detected and was dispersed in the striata of Tg-APP/PS1 mice (Figure 5G,H). Overall, HMGB1 immunoreactivity was detected in both the nuclei and the cytoplasm (double arrows) of microglia in both wildtype (Figure 5E,F) and Tg-APP/PS1 mice (Figure 5G,H). In Iba1-negative cells (the majority of which appeared to be neurons), HMGB1 was also detected in both the nucleus and the cytoplasm (arrowheads in Figure 5E–H). Consistent with this observation, triple-immunofluorescence staining with anti-Aβ, anti-HMGB1, and anti-NeuN antibodies further showed HMGB1 accumulation in both the nuclei and cytoplasm of neurons and NeuN-negative cells in the striata of both wildtype and Tg-APP/PS1 mice (Appendix A). Collectively, these results indicate that, in the striata of Tg-APP/PS1 mice at 9.5 M of age, HMGB1 was located in both the nuclei and the cytoplasm of both neurons and microglia. 

### 3.6. Aβ-Induced Extracellular Secretion of HMGB1 in Neurons

A profound reduction in nuclear HMGB1 in the neurons in the frontal cortices of Tg-APP/PS1 mice at 9.5 M of age prompted us to investigate whether Aβ induces extracellular secretion of HMGB1 from neurons. To generate cytotoxic Aβ oligomers, dissolved Aβ was sonicated and then incubated for 3, 6, 12, or 24 h at 4 °C with shaking (Figure 6A). Incubation of Aβ for 6 h produced oligomeric formation (red box in Figure 6A). When primary cortical cultures were treated with an Aβ oligomer mix (0.1, 0.5, 1, or 3 μM) for 12 or 24 h, HMGB1 significantly accumulated in the culture media when the cells were treated with 1 or 3 μM Aβ for 24 h (Figure 6B,C). ELISA analysis performed on whole-cell lysates and culture media obtained at 24 h after Aβ treatment further confirmed that HMGB1 was significantly increased in both cell lysates and culture media of primary neuron cultures following Aβ treatment (Figure 6D,E). In contrast, treatment with 3 μM of the scrambled Aβ peptide (scAβ) for 24 h did not elevate the HMGB1 levels in cell lysates or its accumulation in the culture media (Figure 6B–E). The cell viability did not change with any dose of Aβ and 3 μM of scAβ (Figure 6F). Taken together, these results indicate that Aβ upregulates HMGB1 production and induces its extracellular secretion in neurons.

### 3.7. Aβ-Induced Extracellular Secretion of HMGB1 in Microglia 

A significant decrease in nuclear HMGB1 in the microglia in the frontal cortices of Tg-APP/PS1 mice at 9.5 M of age led us to investigate the possibility of extracellular secretion of HMGB1 from Aβ-challenged microglia. Treatment of primary microglial cells with Aβ oligomers (0.5, 1, 2, or 3 μM), prepared as described in the experiment shown in Figure 6, caused HMGB1 to accumulate in the culture medium in a dose-dependent manner at both 12 and 24 h after the Aβ treatment (Figure 7A,B). ELISA analysis performed with culture media obtained at 24 h after the Aβ treatment further confirmed the Aβ-mediated HMGB1 secretion from primary microglial cultures (Figure 7C). In contrast, the induction of extracellular secretion of HMGB1 was not detected in microglial cells treated with scAβ (Figure 7A–C). No significant induction of HMGB1 in whole-cell lysates was detected in the ELISA analysis. In addition, the activation of microglia was observed, as evidenced by a significant enhancement of NO production (Figure 7C), but the cell viability did not change with any dose of Aβ or 3 μM or scAβ (Figure 7E). These results indicate that Aβ upregulates HMGB1 production and induces its extracellular secretion by microglia. 

## 4. Discussion

In this study, we demonstrated a significant decrease in HMGB1 protein levels in the frontal cortices of 7.5 M Tg-APP/PS1 mice compared to those in age-matched Wt mice, despite there being almost no changes in HMGB1 protein levels in the striata. Our results showed that, in the frontal cortices of 9.5 M Tg-APP/PS1 mice, nuclear HMGB1 was significantly decreased in both neurons and microglia, and this change was more prominent in neurons, suggesting that HMGB1 is translocated from the nucleus to the cytoplasm and subsequently secreted extracellularly. In addition, the total amount of HMGB1 protein was significantly increased in the sera of aged mice (Figure 1E). Although we did not find direct evidence for the origin of serum HMGB1 in the present study, it is possible that HMGB1 released from neurons and microglia contributed to the increased HMGB1 levels in the sera of aged Tg-APP/PS1 mice. The importance of HMGB1 in the progression of AD’s pathology and the amelioration of AD’s pathology by HMGB1 inhibition have been reported in numerous reports [14,16,17,18,19]. Here, our results demonstrate the cellular and subcellular localization of HMGB1 in Tg-APP/PS1 mice, which can aid our understanding of the role of HMGB1 in the progression of AD’s pathology. In particular, evidence of the translocation of HMGB1 from the nucleus to the cytoplasm of neurons, as revealed by immunohistochemical analysis, strongly supports the notion that HMGB1 is released from neurons during AD’s pathology.

In Tg-APP/PS1 mice at 7.5 M of age, the total amount of HMGB1 in the frontal cortex was significantly lower than that in age-matched wildtype mice, in contrast to that in the striatum, where HMGB1 was detected at similar levels in Tg-APP/PS1 and wildtype control mice. In addition, nuclear HMGB1 levels were significantly reduced in the frontal cortex but not in the striatum of Tg-APP/PS1 mice. It is important to note here that Tg-APP/PS1 mice at 7.5–9.5 M of age showed more Aβ plaques in the frontal cortex than in the striatum [22,29]. The difference in Aβ levels between the frontal cortex and striatum likely alters the distribution of HMGB1 in the nuclei and cytoplasm. In addition, our results suggest that increased release of HMGB1 into the extracellular space (Figure 6 and Figure 7) results in HMGB1 interacting with Aβ, increases neuroinflammation by activating microglia, and promotes Aβ production and/or Aβ accumulation in the frontal cortex. Notably, while some studies have reported elevated levels of HMGB1 in the CSF or sera of AD patients [16,30] or mouse models [31], the total amount of HMGB1 protein in brain tissue—for example, in the hippocampus—is inconsistent; in some studies, the levels were increased [30,32,33], whilst in others there was no difference [34]. In addition, recently, a reduction in HMGB1 levels in the brain due to Tau-induced HMGB1 release was reported in hTau mice [35]. Thus, changes in HMGB1 levels in different brain regions due to AD’s progression need to be studied further.

Our results obtained from immunohistochemistry suggest that HMGB1 translocates from the nuclei to the cytoplasm of neurons in the frontal cortices of Tg-APP/PS1 mice (9.5 M). HMGB1 immunoreactivity was detected in most neurons in the frontal cortices of Tg-APP/PS1 mice, but the levels of cytoplasmic HMGB1 were not enhanced; rather, they appeared to decrease. Our in vitro results support the hypothesis that neuronal HMGB1 is rapidly secreted into the extracellular space after Aβ treatment (Figure 6). Therefore, we speculate that HMGB1 in the nuclei of neurons is translocated to the cytoplasm and secreted extracellularly, and that neuro-derived HMGB1 might constitute a major portion of the extracellular HMGB1. Recently, Fujita et al. (2016) [16] reported Aβ-induced HMGB1 secretion in primary cortical neurons, in which Aβ monomers and oligomers had a higher potency than fibrils. Similar HMGB1 secretion was also observed in primary hippocampal cultures after Aβ treatment [36]. However, the molecular mechanisms by which HMGB1 is secreted by neurons are currently unknown and require further investigation. Furthermore, studying the relative amounts of the two subtypes of HMGB1, disulfide HMGB1 (dsHMGB1) and reduced HMGB1 (reHMGB1), at different times and locations and their differential functions—for example, induction of Aβ oligomerization and inhibition of phagocytosis—may also be important.

Recent studies have reported that extracellular HMGB1 acts as a chaperone for Aβ and inhibits microglial Aβ clearance by interfering with the degradation of Aβ40 and the internalization of Aβ42 by the microglia [15,16,17]. Our finding that microglia form clusters around Aβ plaques in the frontal cortices of Tg-APP/PS1 mice at 9.5 M of age (Figure 4) is consistent with recent reports [14,15]. Additionally, it has been reported that HMGB1 initiates neurite degeneration in the AD brain by phosphorylating TLR4-myristoylated alanine-rich C-kinase substrate (MARCKS) and induces Aβ oligomers and protofibrils, exacerbating AD’s pathology [16]. Moreover, HMGB1 mediates the BBB dysfunction through disruption of zona ocludin-1, causing transendothelial migration of WBCs and neuroinflammation in AD [37]. It is notable that TLR4 and RAGE played critical roles in the HMGB1-mediated pathology of AD in the abovementioned reports [16,36]. Moreover, loss of long-term memory, a clinical symptom in AD, was further aggravated by intracerebroventricular injection of HMGB1; however, this effect was blocked by TLR4 antagonist injection in RAGE-deficient mice, indicating that HMGB1-induced memory dysfunction is TLR4- and RAGE-dependent [18]. HMGB1 released from neurons might affect AD’s pathology at multiple stages during the progression of AD, by forming a vicious cycle that exacerbates the pathology. Moreover, since neuroinflammation is an important contributor to AD’s pathology [5] and HMGB1 functions as a pro-inflammatory cytokine [38,39], HMGB1 also contributes to AD’s progression by priming neurodegeneration [6]. 

HMGB1 is a known molecular signature of senescent cells and is actively released from senescent cells into the extracellular milieu [40]. HMGB1 also upregulates the expression of inflammatory cytokines and plays a role in neuroinflammatory priming [6,40,41,42]. In the present study, we showed that HMGB1 protein levels were significantly elevated in brain tissue homogenates, as well as in the sera of aged wildtype mice. In the frontal cortices of aged wildtype mice, the overall intensity of HMGB1 immunoreactivity decreased, and these changes were more pronounced in microglia than in neurons, indicating that microglia may be the dominant cell type secreting HMGB1 in the brains of aged mice. These results are consistent with those of Fonken et al. [6], who showed that increased HMGB1 secretion occurs mainly in microglia in the CA1 and DG regions of the hippocampus in aged mice (24 M) and mediates aging-induced neuroinflammatory priming. In addition, HMGB1 levels are changed in a cell-type-specific manner during aging (24 M): it is gradually reduced in neurons but it increases in astrocytes [43]. Furthermore, blocking HMGB1’s function in senescent astrocytes using an anti-HMGB1 antibody prevents paracrine senescence in adjacent healthy astrocytes [35]. The contribution of senescent glial cells to the initiation and progression of neurodegeneration and cognitive dysfunction in AD and FTD [44,45] has recently attracted attention, although their relationship with HMGB1 requires further study. Despite the possible importance of microglia-derived HMGB1 in AD-like brains, our results support the hypothesis that both neuron-derived HMGB1 and microglia-derived HMGB1 play important roles in AD’s pathology.

## Figures and Tables

**Figure 1 cells-13-00189-f001:**
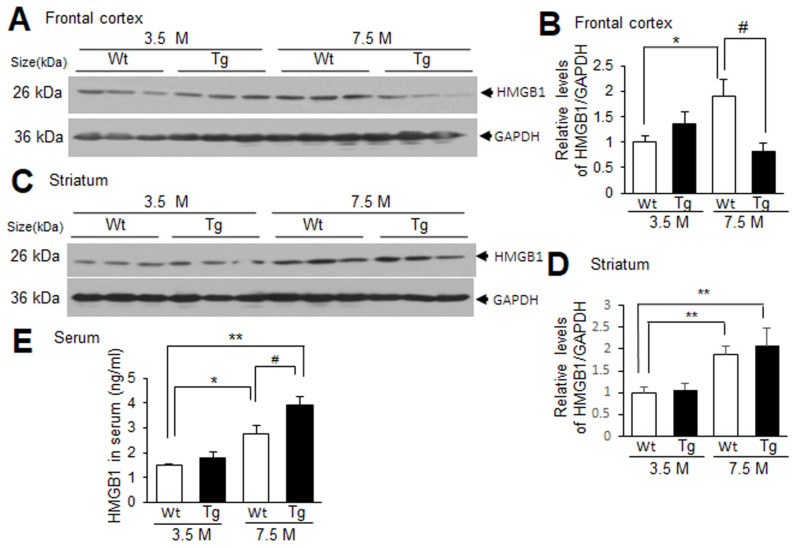
HMGB1 protein levels in the frontal cortex, striatum, and serum of 3.5 M or 7.5 M Tg-APP/PS1 mice and age-matched control mice: (**A**–**D**) Protein samples were prepared from the frontal cortices (**A**,**B**) and striata (**C**,**D**) of Tg-APP/PS1 mice at 3.5 or 7.5 M of age or age-matched wildtype (Wt) mice, and HMGB1 protein levels were determined by immunoblotting. (**E**) Protein samples were prepared from the sera of Tg-APP/PS1 or age-matched Wt mice at 3.5 or 7.5 M of age, and HMGB1 protein levels were determined by ELISA. Representative immunoblots are presented in (**A**,**C**), and the quantification results are presented as means ± SEMs (n = 6 for (**B**,**D**), n = 3 for (**E**)); * *p* < 0.05, ** *p* < 0.01 vs. 3.5 M Wt, # *p* < 0.05 between indicated groups.

**Figure 2 cells-13-00189-f002:**
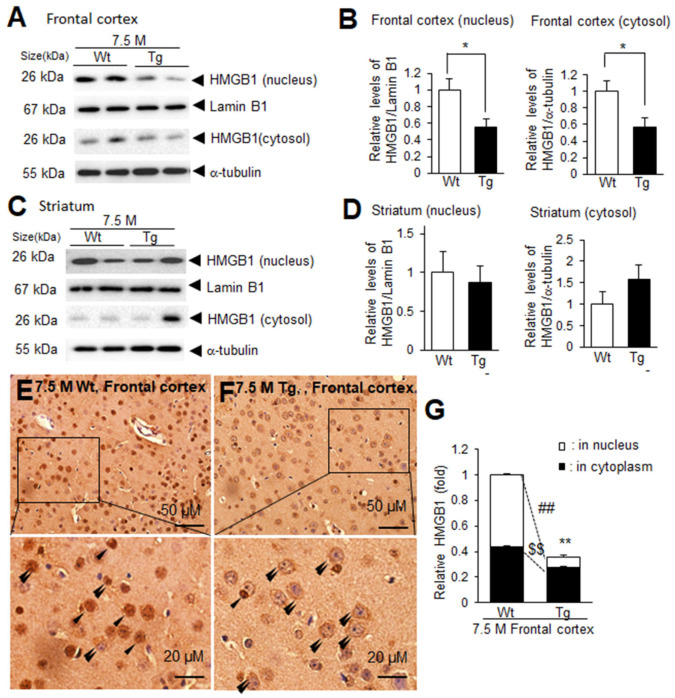
Nuclear and cytosolic HMGB1 protein levels in the frontal cortices and striata of 7.5 M Tg-APP/PS1 and age-matched control mice: Protein samples were prepared from the parietal cortices (**A**,**B**) and striata (**C**,**D**) of 7.5 M Tg or age-matched Wt mice, and the levels of HMGB1 protein in the nuclei and cytoplasm were examined using immunoblotting. Representative immunoblots are presented in (**A**,**C**), and the quantification results are presented as means ± SEMs (n = 4); * *p* < 0.05 vs. 7.5 M Wt. (**E**–**G**) Immunohistochemistry was performed using an anti-HMGB1 antibody, followed by counterstaining with H&E. Representative images are shown in (**E**) (Wt) and (**F**) (Tg), and the quantified results of nuclear and cytoplasmic HMGB1 obtained using ImageJ software -v. 1.50 (NIH, Bethesda, MD, USA) are presented as means ± SEMs (n = 12 from 3 animals) in (**G**). Arrowheads and double arrowheads indicate HMGB1 immunoreactivity in the nucleus or in the cytoplasm, respectively. The scale bars in (**E**,**F**) represent 50 µm, and those in the high-magnification images represent 20 µm; ** *p* < 0.01 vs. 7.5 M Wt (total amounts), ## *p* < 0.01 vs. 7.5 M Wt (nuclear HMGB1), ^$$^
*p* < 0.01 vs. 7.5 M Wt (cytoplasmic HMGB1).

**Figure 3 cells-13-00189-f003:**
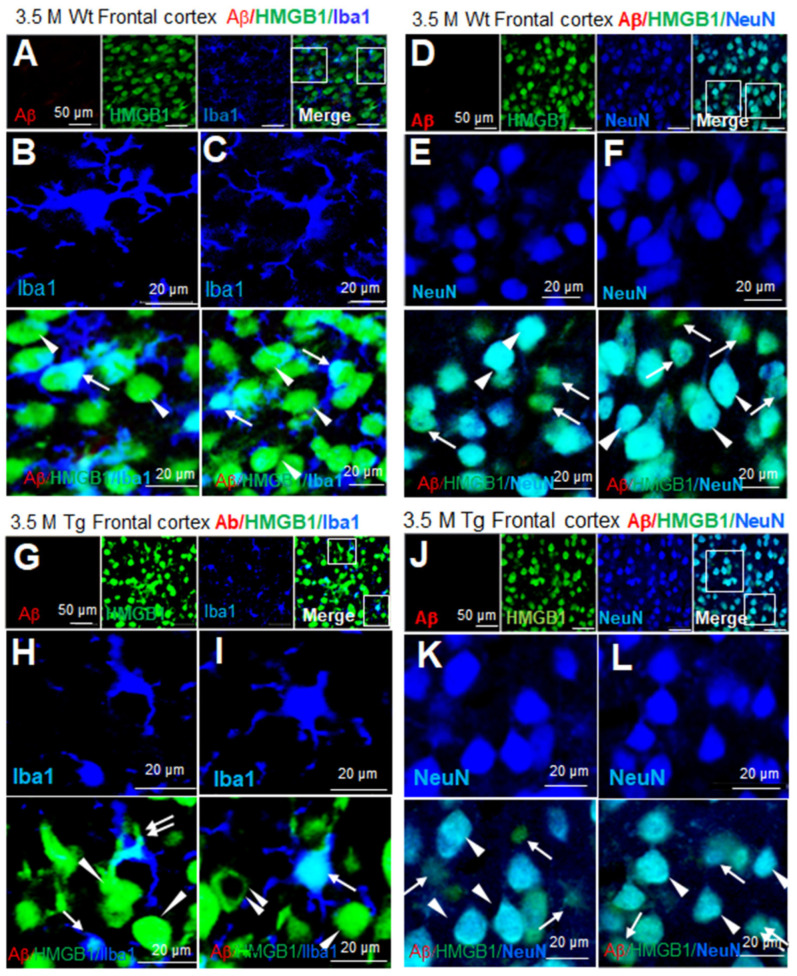
Cellular and subcellular localization of HMGB1 in the frontal cortices of 3.5 M Tg-APP/PS1 and age-matched control mice: Coronal brain sections were prepared from 3.5 M Tg (**G**–**L**) and age-matched control (**A**–**F**) mice, and triple-immunofluorescence staining was conducted with anti-Aβ, anti-HMGB1, and anti-Iba1 (**A**–**C**,**G**–**I**) or anti-Aβ, anti-HMGB1, and anti-NeuN (**D**–**F**,**J**–**L**) antibodies. Arrows indicate anti-HMGB1 immunoreactivity in microglia (**B**,**C**,**H**,**I**) or NeuN-negative cells (**E**,**F**,**K**,**L**). Arrowheads indicate anti-HMGB1 immunoreactivity in neurons (**E**,**F**,**K**,**L**) or Iba1-negative cells (**H**,**I**). The double arrow in (**H**) indicates anti-HMGB1 immunoreactivity in the cytoplasm of microglia, and the double arrowhead in (**I**) indicates anti-HMGB1 immunoreactivity in the cytoplasm of Iba1-negative cells (neurons). Scale bars represent 20 or 50 µm.

**Figure 4 cells-13-00189-f004:**
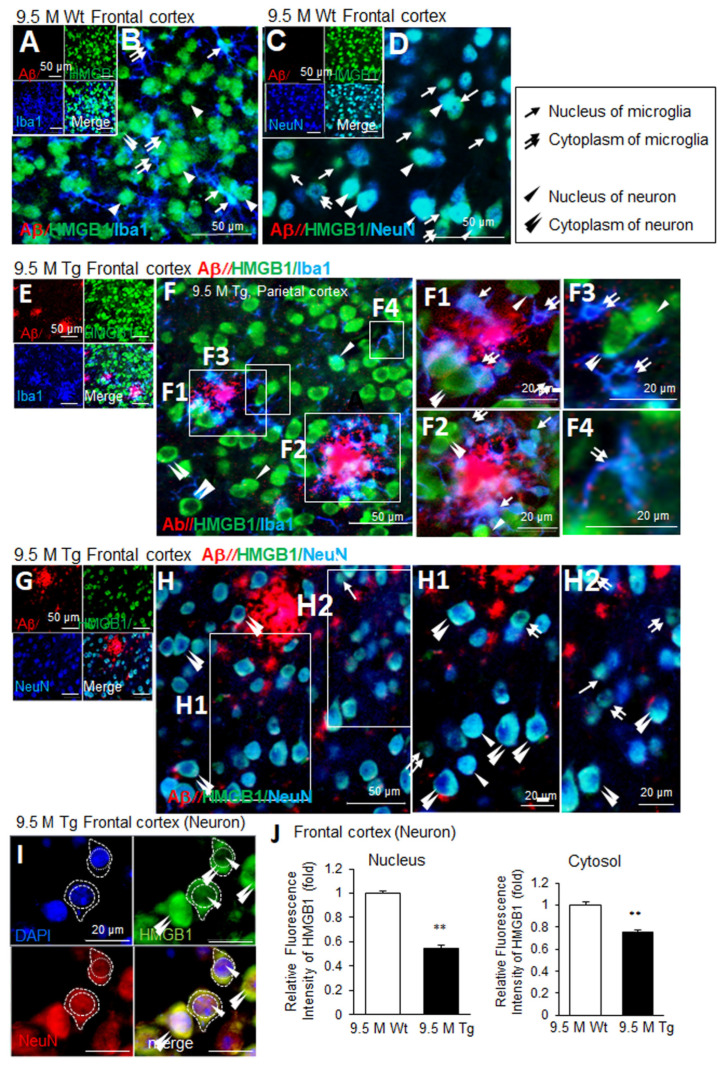
Cellular and subcellular localization of HMGB1 in the frontal cortices of 9.5 M Tg-APP/PS1 and age-matched control mice: Coronal brain sections were prepared from 9.5 M Tg (**E**–**H**) and age-matched control (**A**–**D**) mice. Triple-immunofluorescence staining was performed using anti-Aβ, anti-HMGB1, and anti-Iba1 (**A**,**B**,**E**,**F**) or anti-Aβ, anti-HMGB1, and anti-NeuN (**C**,**D**,**G**,**H**) antibodies. (**I**,**J**) Triple-immunofluorescence staining was performed using anti-HMGB1 antibodies, anti-NeuN antibodies, and DAPI. HMGB1 immunoreactivity (fluorescence intensity), co-localized with DAPI (nuclear) or not (cytoplasmic), was assessed in individual neurons. Staining intensity was measured using ImageJ software -v. 1.50, and the quantified results are presented as means ± SEMs (n = 30). Arrows indicate the anti-HMGB1 immunoreactivity in microglia (**B**,**F1**–**F4**) and NeuN-negative cells (**D**,**H2**). Arrowheads indicate anti-HMGB1 immunoreactivity in neurons (**D**,**H1**) or Iba1-negative cells (**B**,**F1**–**F3**). Double arrows indicate the anti-HMGB1 immunoreactivity in the cytosol of microglia (**B**,**F1**–**F4**) or NeuN-negative cells (**D**,**H1**,**H2**). Double arrowheads indicate the anti-HMGB1 immunoreactivity in the cytosol of neurons (**H1**–**H2**) or Iba1-negative cells (**B**,**F1**–**F3**). Scale bars represent 20 or 50 µm. ** *p* < 0.01 vs. 9.5 M Wt (nuclear HMGB1 or cytoplasmic HMGB1).

**Figure 5 cells-13-00189-f005:**
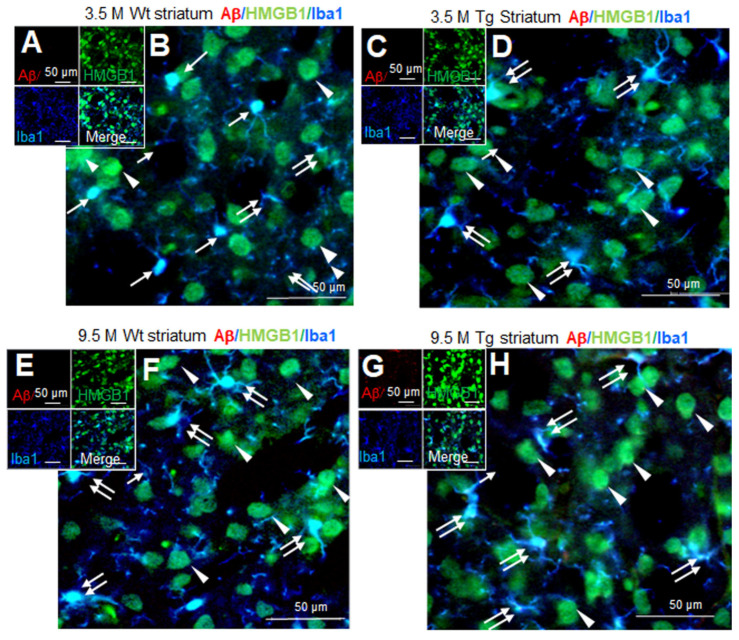
Cellular and subcellular localization of HMGB1 in the striata of 3.5 and 9.5 M Tg-APP/PS1 and age-matched control mice: Coronal brain sections were prepared from 3.5 M (**C**,**D**) or 9.5 M Tg (**G**,**H**) and age-matched control (**A**,**B**,**E**,**F**) mice. Triple-immunofluorescence staining was performed using anti-Aβ, anti-HMGB1, and anti-Iba1 (**A**–**H**) antibodies. Arrows indicate the anti-HMGB1 immunoreactivity in microglia (**B**,**D**,**F**,**H**). Arrowheads indicate the anti-HMGB1 immunoreactivity in Iba1-negative cells (**B**,**D**,**F**,**H**). Double arrows indicate the anti-HMGB1 immunoreactivity in the cytosol of microglia (**B**,**D**,**F**,**H**). Scale bars represent 20 or 50 µm.

**Figure 6 cells-13-00189-f006:**
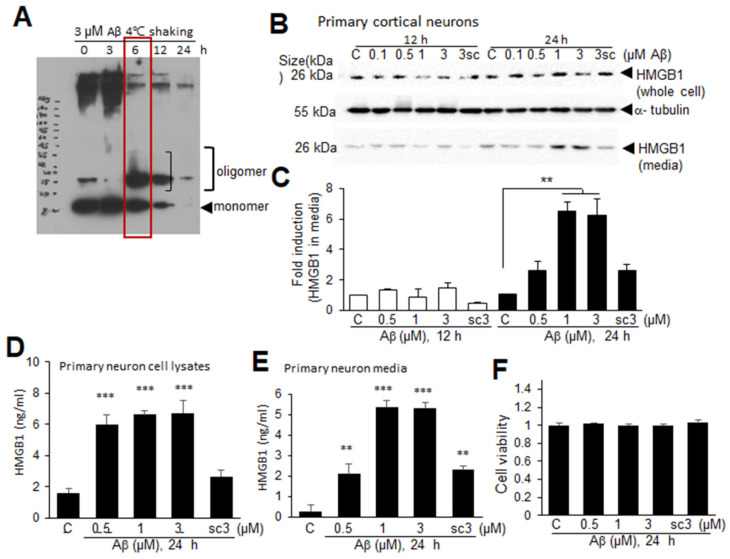
Aβ-induced HMGB1 secretion in primary cortical cultures: (**A**) Aβ or scrambled Aβ (scAβ) was sonicated for 1 min using a 50/60 Hz sonicator to remove any pre-aggregates, and then incubated with rotation for 3, 6, 12, or 24 h at 4 °C. Oligomer formation was examined using SDS gels. (**B**,**C**) Primary cortical cultures were treated with Aβ (0.1, 0.5, 1, or 3 μM, 6 h rotation) or scrambled Aβ (scAβ, 3 μM, 6 h rotation as indicated in red box) for 12 or 24 h, and the levels of HMGB1 in the whole-cell lysates and media were examined using immunoblotting. Representative immunoblots are presented in (**B**), and the quantification results are presented as the means ± SEMs (n = 4) in (**C**). (**D**–**F**) Primary cortical cultures were treated with Aβ (0.5, 1, or 3 μM) or scrambled Aβ (sc, 3 μM) for 24 h, the levels of HMGB1 in the whole-cell lysates and media were examined using ELISA (**D**,**E**), and cell viability was accessed using an MTT assay (**F**). The quantification results are presented as the means ± SEM (n = 3); ** *p* < 0.01, *** *p* < 0.001 vs. PBS-treated control cells.

**Figure 7 cells-13-00189-f007:**
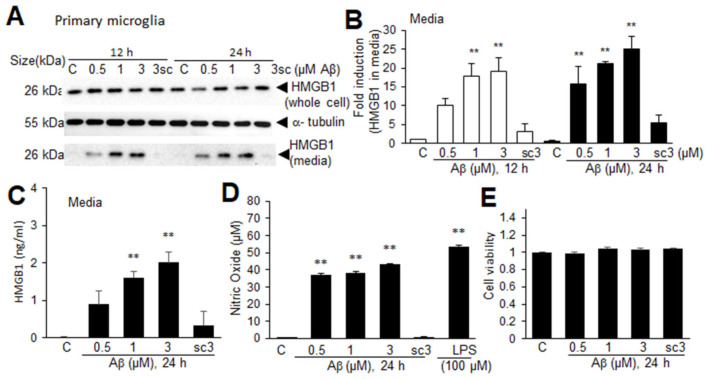
Aβ-induced HMGB1 secretion in primary microglial cultures: (**A**,**B**) Primary microglial cultures were treated with Aβ (0.5, 1, or 3 μM) or scrambled Aβ (scAβ, 3 μM) for 12 or 24 h, and the levels of HMGB1 in the whole-cell lysates and media were examined using immunoblotting. Representative immunoblots are presented in (**A**), and the quantification results of the media are presented as the means ± SEMs (n = 4) in (**B**). (**C**–**E**) Primary microglial cultures were treated with Aβ (0.5, 1, 3 μM) or scrambled Aβ (3 μM) for 24 h, and the HMGB1 levels in the culture media (**C**), NO levels (**D**), and cell viability (**E**) were assessed using ELISA, the Griess assay, and the MTT assay, respectively. The quantification results are presented as the means ± SEMs (n = 3); ** *p* < 0.01 vs. PBS-treated control cells.

## Data Availability

Data are contained within the article and Appendix A.

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
