# Peer review of "Age-Dependent and Aβ-Induced Dynamic Changes in the Subcellular Localization of HMGB1 in Neurons and Microglia in the Brains of an Animal Model of Alzheimer’s Disease"

_cells, 2024, doi:10.3390/cells13020189_

Round 1
Reviewer 1 Report
Comments and Suggestions for Authors
Major
1. In the introduction, authors speak about TLR4 and RAGE receptors but provide no insight into the mechanistic pathway these are part of and how they are related to HMGB1. Authors should elaborate on the relationship between these molecular targets.
2. In the introduction, authors mention several animal models including Tg2576, Tg6799 and ultimately Tg-APP/PS1 which was used for experiments in this study. However, the authors do not provide any details on these mouse models, how they differ from each other and why they selected the Tg-APP/PS1 mouse for this particular study. The authors must provide details on all mouse models mentioned and justify why they selected the Tg-APP/PS1 mouse model for their study. How is the pathological progression of disease in Tg-APP/PS1 in comparison to human pathological progression? Authors should cite prior literature on plaque formation observed in the Tg-APP/PS1 mouse in various brain structures.
3. The authors have conducted a nitric oxide assay in primary microglia however there is no sufficient rationale for these experiments; authors must provide more physiological context and elaborate on why they selected this assay for microglial activation.
4. In Figure 2 A,B, the immunoblots of cytoplasmic fractions of HMGB1 do not convincingly indicate that the levels are reduced in 7.5M Tg mice versus Wt frontal cortex. Can the authors include ELISA data for the nuclear and cytoplasmic factions which is more quantitative than the immunoblot ? If ELISA is not possible, a more representative immunoblot of the cytoplasmic fraction must be presented to conclude that there is reduced HMGB1 in Tg versus Wt mice cytoplasm in the frontal cortex.
5. Authors should explain the method for generating and validating AB cytotoxic oligomers and whether this is a novel method. If it is novel, how did the authors validate the AB oligomer effect besides just the HGMB1 levels? If it is a method previously used by others, authors should cite the relevant literature.
6. In Figure 6, why can't ELISA for HMGB1 be used as an assay to quantify levels. Immunoblotting for HMGB1 is semi-quantitative at best. Also, tubulin is an intracellular marker. For the extracellular secreted HMGB1, a loading control that is extracellular is preferred for normalization.
7. From the immunoblots (Fig 6 and 7), it is not clear that the HMGB1 levels are increased in whole cell lysates for either primary cortical neurons or primary microglia with AB oligomer treatments. Authors should include better immunoblots or supporting ELISA data to show conclude that total HMGB1 levels increase with AB oligomer treatments in these primary cell models.
8. Authors must discuss the function of HMGB1 in relation to the blood brain barrier as it explains disease progression and proinflammatory effects of HMGB1 on BBB integrity (See https://jneuroinflammation.biomedcentral.com/articles/10.1186/s12974-016-0670-z)
Minor
1. On line 61, authors perhaps intended to state "...injection of HMGB1 exacerbates AB42-induced long term memory loss." The word "loss" is missing in the statement.
2. Line 381 should likely say. "....secretion of HMGB1.." and not "secretion of AB"?
3. Line 470, authors cite 'pharmacological inhibition of microglial senescence.." but do not elaborate on what this pathway is while citing this paper. The relevance is therefore lost. Authors should elaborate on why they cited this piece of literature.
Overall: It is unclear that the total HMGB1 levels in whole cell lysates are increasing with AB oligomer treatments. Results are not convincing. Better blots and ELISA data must be included here.
Comments on the Quality of English Language
Quality of English language is fine but some phrases are incomplete or erroneous. Can be edited.
Author Response
Reviewer 1
Major
- In the introduction, authors speak about TLR4 and RAGE receptors but provide no insight into the mechanistic pathway these are part of and how they are related to HMGB1. Authors should elaborate on the relationship between these molecular targets.
Response: We agree with the reviewer’s point. We addressed the importance of TLR4 and RAGE receptors in AD pathology in relation with HMGB1 function in the Discussion section of the revised manuscript. We also provided appropriate references for each function.
- In the introduction, authors mention several animal models including Tg2576, Tg6799 and ultimately Tg-APP/PS1 which was used for experiments in this study. However, the authors do not provide any details on these mouse models, how they differ from each other and why they selected the Tg-APP/PS1 mouse for this particular study. The authors must provide details on all mouse models mentioned and justify why they selected the Tg-APP/PS1 mouse model for their study. How is the pathological progression of disease in Tg-APP/PS1 in comparison to human pathological progression? Authors should cite prior literature on plaque formation observed in the Tg-APP/PS1 mouse in various brain structures.
Response: Thanks. We have added and revised the details of Tg2576, Tg6799, and Tg-APP/PS1 in the revised manuscript. Tg2576 mice overexpress a mutant form of APP (isoform 695) with the Swedish mutation (KM670/671NL) under the control of the PDGF promoter. The Tg6799 mouse line is more commonly known as the 5xFAD mouse. We have renamed it accordingly in the manuscript. 5xFAD mice express human APP and PSEN1 transgenes with a total of five AD-linked mutations: the Swedish (K670N/M671L), Florida (I716V), and London (V717I) mutations in APP, and the M146L and L286V mutations in PSEN1 under the control of the Thy-1 promoter. Tg-APP/PS1 mice express the human Swedish amyloid precursor protein (APPswe) and the ΔE9 presenilin 1 mutation (PSEN1dE9) under the control of the PDGF promoter.
All three mouse lines (Tg2576, 5xFAD, and Tg-APP/PS1) show age-dependent accumulation of plaque deposition and Ab42-related neuropathologies (Lee and Han, 2013, Exp Neurobiol. 22(2):84-95): (i) Tg2576 mice show plaque deposition starting from approximately 10 months of age. (ii) 5xFAD mice show plaque deposition starting from approximately 2.5-3 months of age. (iii) Tg-APP/PS1 mice show plaque deposition starting from approximately 6.5 months of age.
Despite limitations in recapitulating human AD pathology, these three models have been extensively used as animal models in studying for human AD over the past decades. In recent years, many laboratories have preferred 5xFAD and Tg-APP/PS1 mice over Tg2576, primarily due to the early plaque deposition observed in 5xFAD mice and the relatively early plaque deposition and age-dependent changes observed in Tg-APP/PS1 mice. We chose the Tg-APP/PS1 mouse line for the present study because it presents nearly all aspects of Ab42-related pathologies known in animal models.
We added and revised the details of Tg-APP/PS1 in the last paragraph of the Introduction section of the revised manuscript.
- The authors have conducted a nitric oxide assay in primary microglia however there is no sufficient rationale for these experiments; authors must provide more physiological context and elaborate on why they selected this assay for microglial activation.
Response: Nitric oxide (NO) release is a well-characterized marker of microglial activation. Microglia, like BV2 cells, exhibit macrophage-like properties. Therefore, when exposed to pro-inflammatory factors, they release NO as a consequence of the activation of the canonical signaling cascade, leading to iNOS upregulation. Accordingly, the nitric oxide synthase (NOS) assay is frequently used and is a reliable marker for the state of activated microglia. As demonstrated in Figure 7, HMGB1 release is proportional to the level of microglial activation or NO release. We mentioned more clearly this rationale in the Results section of the revised manuscript.
- In Figure 2 A,B, the immunoblots of cytoplasmic fractions of HMGB1 do not convincingly indicate that the levels are reduced in 7.5M Tg mice versus Wt frontal cortex. Can the authors include ELISA data for the nuclear and cytoplasmic factions which is more quantitative than the immunoblot ? If ELISA is not possible, a more representative immunoblot of the cytoplasmic fraction must be presented to conclude that there is reduced HMGB1 in Tg versus Wt mice cytoplasm in the frontal cortex.
Response: We conducted additional experiments to confirm immunoblotting results. Immunohistochemistry using anti-HMGB1 antibody, followed by H&E staining, revealed a significant decrease in HMGB1 immunoreactivity in the frontal cortices of 7.5 M Tg mice. Notably, there was a marked alteration in the subcellular localization of HMGB1, which was primarily detected in the cytoplasm (as indicated by double arrowheads in Figs. 6E and F of the revised manuscript). These results indicate that HMGB1 levels were reduced in both nucleus and cytoplasm in the frontal cortices. We have incorporated these new data sets in Figures 6E and F of the revised manuscript and have addressed them in the Results section.
- Authors should explain the method for generating and validating AB cytotoxic oligomers and whether this is a novel method. If it is novel, how did the authors validate the AB oligomer effect besides just the HGMB1 levels? If it is a method previously used by others, authors should cite the relevant literature.
Response: We cited the two papers that we referred to in the revised manuscript.
- Ahmed, M.; Davis, J.; Aucoin, D.; Sato, T.; Ahuja, S.; Aimoto, D.; Elliott, J.; Van Nostrand, W.E.; Smith, S.O. Structural conversion of neurotoxic amyloid-β(1–42) oligomers to fibrils. Nat. Struct. Mo.l Biol. 2010, 17, 561–567.
- Itkin, A.; Dupres, V.; Dufrene, Y.F.; Bechinger, B.; Ruysschaert, J-M.; Raussens, V. Calcium Ions Promote Formation of Amyloid b-Peptide (1–40) Oligomers Causally Implicated in Neuronal Toxicity of Alzheimer’s Disease. PLoS ONE. 2011, 6, e18250.
- In Figure 6, why can't ELISA for HMGB1 be used as an assay to quantify levels. Immunoblotting for HMGB1 is semi-quantitative at best. Also, tubulin is an intracellular marker. For the extracellular secreted HMGB1, a loading control that is extracellular is preferred for normalization.
Response: By following reviewer’s suggestion, we performed ELISA on cell lysates and culture media of primary cortical culture and primary microglia cultures. Results provide clear confirmation of Ab-mediated extracellular secretion of HMGB1.We have incorporated new ELISA data in Figures 6D, 6E, and 7C of the Results section and have addressed them in the revised manuscript.
- From the immunoblots (Fig 6 and 7), it is not clear that the HMGB1 levels are increased in whole cell lysates for either primary cortical neurons or primary microglia with AB oligomer treatments. Authors should include better immunoblots or supporting ELISA data to show conclude that total HMGB1 levels increase with AB oligomer treatments in these primary cell models.
Response: By following reviewer’s suggestion, we performed ELISA on cell lysates and found that HMGB1 levels in cell lysates were significantly increased by Ab in primary cortical cultures, but not in primary microglia. We have included the new ELISA data in Figure 6D and Supplementary figure 3 of the revised manuscript.
- Authors must discuss the function of HMGB1 in relation to the blood brain barrier as it explains disease progression and proinflammatory effects of HMGB1 on BBB integrity (See https://jneuroinflammation.biomedcentral.com/articles/10.1186/s12974-016-0670-z)
Response: By following reviewer’s suggestion, we addressed HMGB1 function in relation to the BBB disruption during the disease progression of AD in the Discussion section of the revised manuscript. We also added a new reference in the revised manuscript.
Minor
- On line 61, authors perhaps intended to state "...injection of HMGB1 exacerbates AB42-induced long term memory loss." The word "loss" is missing in the statement.
Response: We corrected it.
- Line 381 should likely say. "....secretion of HMGB1.." and not "secretion of AB"?
Response: We corrected it.
- Line 470, authors cite 'pharmacological inhibition of microglial senescence.." but do not elaborate on what this pathway is while citing this paper. The relevance is therefore lost. Authors should elaborate on why they cited this piece of literature.
Response: Hu et al (2021) claimed that in Alzheimer’s-like pathology, a fraction of microglia undergo replicative senescence. In the above-mentioned report, authors showed that prevention of microglia proliferation using CSF1R inhibitor (GW2580) impairs the development of microglial senescence, causing reduced amyloidosis. However, since is not directly related with HMGB1, we deleted this sentence and related reference in the revised manuscript.
Reviewer 2 Report
Comments and Suggestions for Authors
Please see the attached file.

Author Response
- It is unclear that the total HMGB1 levels in whole cell lysates are increasing with AB oligomer treatments. Results are not convincing. Better blots and ELISA data must be included here.
Response: By following reviewer’s suggestion, we performed ELISA on cell lysates and found that HMGB1 levels in cell lysates were significantly increased by Ab in primary cortical cultures, but not in primary microglia. We have included the new ELISA data in Figure 6D and Supplementary figure 3 of the revised manuscript.
- Authors describe the modification in HMGB1 intracellular localization in APP/PS1 transgenic mice. In vivoand in vitroresults are clearly shown. However, immunofluorescence assays should be measured and accompanied by numeric results in graphs.
Response: To further quantify HMGB1’s subcellular localization, we performed additional experiments. We measured the fluorescent intensity of HMGB1 co-localized with DAPI (nuclear) or not (cytoplasmic) in neurons of the frontal cortices of 9.5 M Tg mice. This analysis revealed a significant decrease in nuclear HMGB1 (arrowheads), while cytoplasmic HMGB1 (double arrowheads) showed a slight but significant decrease. We have incorporated these new data sets in Figures 4I and 4J of the revised manuscript and have addressed them in the Results section.
- Authors performed in vitroanalysis with WT cells (neurons and microglia) and exogenous Abeta peptide. I missed the analysis of transgenic primary neurons and microglia (maybe astrocytes too) that produce Abeta peptide
Response: I agree reviewer’s point. We plan to do additional experiments including those using transgenic cells, which will provide deeper insights into the issue.
- Line 441 A instead of Aβ
Response: We corrected it.
Reviewer 3 Report
Comments and Suggestions for Authors
This experimental study addresses the cell dynamics of HMBG1, ie an intrinsic DAMP molecule that activates microglia and neuroinflammation, in the brain of a murine model of Alzheimer’s disease as a function of age and brain region. The study uses a battery of complementary approaches, both ex-vivo, in situ and in vitro, following a highly pertinent wire. The methods are well conducted and yielded a coherent set of positive original results. Figures and well composed and results descriptions are very well written. Discussion is straightforward and pertinent.
However, several minor corrections should be brought to the manuscript before it be suitable for publication, as listed below.
Scientific issues
Lines 18-19: The writing of this sentence must be alleviated and clarified.
L. 69-70 : Please authors add one sentence to justify the choice of frontal cortex and striatum as the structures to be addressed in their present study. All also another sentence (either here in Introduction, or in the first paragraph of Materials & Methods below) to justify the ages chosen for your assays.
L. 82: The type of extracts for genotyping the mice must be mentioned here (DNA, of which organ?).
L. 105: The parameters for electrophoresis and electrotransfer (including the membrane used) must be added here.
L. 256: Clarify the description of HMBG1 immunolabelling: “large cell size” is indeed repeated twice about neuronal contents while the nuclear labeling is confuse.
Minor details
L. 151: Add “of” after “per 24-well”.
L. 162: Change “seed” into “seeded”.
Comments on the Quality of English Language
This manuscript is globally well written.
Author Response
Scientific issues Lines 18-19: The writing of this sentence must be alleviated and clarified.
Response: We clarified the sentence by mentioning frontal cortex and striatum in the revised manuscript.
- 69-70: Please authors add one sentence to justify the choice of frontal cortex and striatum as the structures to be addressed in their present study. All also another sentence (either here in Introduction, or in the first paragraph of Materials & Methods below) to justify the ages chosen for your assays.
Response: We mentioned the reason to choose frontal cortex and striatum in the Discussion section: “Tg-APP/PS1 mice at 7.5 – 9.5 M of age showed more Ab plaques in the frontal cortex than in the striatum [22,27]”. We addressed this as well as the reason to choose 7.5~9.5 M in the first paragraph of the Results section (3.1) in the revised manuscript.
- 82: The type of extracts for genotyping the mice must be mentioned here (DNA, of which organ?).
Response: Genotyping was conducted using genomic PCR of tail biopsy samples. We included this information in the Methods section (2.1) of the revised manuscript.
- 105: The parameters for electrophoresis and electrotransfer (including the membrane used) must be added here.
Response: We have included the parameters of electrophoresis and electrotransfer in the Results section of the revised manuscript.
- 256: Clarify the description of HMBG1 immunolabelling: “large cell size” is indeed repeated twice about neuronal contents while the nuclear labeling is confuse.
Response: For better differentiation between nuclear and cytoplasmic HMGB1, we conducted additional sets of triple fluorescence staining using anti-HMGB1 antibody, anti-NeuN antibody, and DAPI. We measured the fluorescence intensity of HMGB1 co-localized with DAPI (nucleus) or not (cytoplasm) in neurons in frontal cortices of 9.5 M Tg mice. This analysis demonstrated a significantly decrease in nuclear HMGB1 (arrowheads), while cytoplasmic HMGB1 (double arrowheads) exhibited a slightbut significant reduction. We have incorporated these new data sets in Figures 4I and 4J of the revised manuscript and addressed them in the Results section.
Minor details
- 151: Add “of” after “per 24-well”.
Response: We corrected it.
- 162: Change “seed” into “seeded”.
Response: We corrected it.
Reviewer 4 Report
Comments and Suggestions for Authors
The study is interesting but appears to have both limited novelty and impact for the field of AD research.
Major limitations:
Regarding WB analyses performed in frontal cortex and in the striatum (Figure 1), HMGB1 levels are expressed as relative values with respect to WT 3.5M group (which is equal to 1 in the figure). However, maintaining this group as reference level for all the experimental groups represents a confounding factor. I would suggest to show densitometric quantification of HMGB1 levels as HMGB1/GAPDH ratio for each experimental group, and analyze group differences moving from this values. A similar approach should be applied also in other WB analyses (Figure 2).
Why were the global levels and the subcellular localization of HMGB1 investigated through WB in 7.5 M Tg-APP/PS1 mice and through immunofluorescence in 9.5 M Tg-APP/PS1 mice? Why the Authors selected a different time-point for such analyses?
Minor:
The Authors should better describe the ELISA methods (kit that has been utilized, sensibility of the assay).
Author Response
Major limitations:
- Regarding WB analyses performed in frontal cortex and in the striatum (Figure 1), HMGB1 levels are expressed as relative values with respect to WT 3.5M group (which is equal to 1 in the figure). However, maintaining this group as reference level for all the experimental groups represents a confounding factor. I would suggest to show densitometric quantification of HMGB1 levels as HMGB1/GAPDH ratio for each experimental group, and analyze group differences moving from this values. A similar approach should be applied also in other WB analyses (Figure 2).
Response: We understand reviewer’s concern. However, we analyzed all data as exactly the reviewer suggested. We performed densitometric quantification of HMGB1 levels as HMGB1/GAPDH ratio for each experimental group and analyze group differences from this values. We changed index of y-axis of Figs. 1B and 1D to “Relative levels of HMGB1/GAPDH”, or of Figs. 2B and 2D to “Relative levels of HMGB1/a-tubulin”, in the revised manuscript.
- Why were the global levels and the subcellular localization of HMGB1 investigated through WB in 7.5 M Tg-APP/PS1 mice and through immunofluorescence in 9.5 M Tg-APP/PS1 mice? Why the Authors selected a different time-point for such analyses?
Response: Thank you for your comment. As demonstrated, the change in the subcellular localization of HMGB1 in 7.5-M-old Tg-APP/PS1 mice was clearly detected by the Western blot method. As Tg-APP/PS1 mice typically exhibit progressive and severe pathology with age, our finding of Aβ42-dependent HMGB1 release led us to predict that immunohistochemical detection of HMGB1 localization in adult brain cells should be more apparent in brains with more advanced pathology. Therefore, we chose to analyze 9.5-month-old Tg-APP/PS1 mice, which usually exhibit greater Ab42-related pathology compared to 7.5-month-old mice. We did not include immunohistochemical data for 7.5-month-old Tg-APP/PS1 mice in Figure 4, partly due to the limited availability of Tg mice.
Minor:
- The Authors should better describe the ELISA methods (kit that has been utilized, sensibility of the assay).
Response: By following reviewer’s suggestion, we included a new paragraph for ELISA (2.4) in the Materials and Methods section of the revised manuscript.
Round 2
Reviewer 1 Report
Comments and Suggestions for Authors
The authors have adequately addressed all my comments from Review 1. This is an interesting study that further explores HMGB1 in Alzheimer's disease pathology.
Reviewer 2 Report
Comments and Suggestions for Authors
The manuscript entitled “Age-Dependent and Aβ-Induced Dynamic Changes in Subcellular Localization of HMGB1 in Neurons and Microglia in the 3 Brain of an Animal Model of Alzheimer’s Disease” has been improved and in my opinion is ready to be published in Cells
Reviewer 3 Report
Comments and Suggestions for Authors
Authors have satisfactorily addressed thé issues that had been raised. This hard work IS now ready tombé published.